# Decoder Denoising Pretraining for Semantic Segmentation

**Emmanuel Asiedu Brempong**[†]                    *brempong@google.com*
*Google Research*

**Simon Kornblith**                    *skornblith@google.com*
*Google Research*

**Ting Chen**                    *iamtingchen@google.com*
*Google Research*

**Niki Parmar**                    *nikip@google.com*
*Google Research*

**Matthias Minderer**[*]                    *mjlm@google.com*
*Google Research*

**Mohammad Norouzi**[*]                    *mnorouzi@google.com*
*Google Research*

**Reviewed on OpenReview:** *https: // openreview. net/ forum? id=D3WIOQG7dC*

## Abstract

Semantic segmentation labels are expensive and time consuming to acquire. Hence, pretraining is commonly used to improve the label-efficiency of segmentation models. Typically, the encoder of a segmentation model is pretrained as a classifier and the decoder is randomly initialized. Here, we argue that random initialization of the decoder can be suboptimal, especially when few labeled examples are available. We propose a decoder pretraining approach based on denoising, which can be combined with supervised pretraining of the encoder. We find that decoder denoising pretraining on the ImageNet dataset strongly outperforms encoder-only supervised pretraining. Despite its simplicity, decoder denoising pretraining achieves state-of-the-art results on label-efficient semantic segmentation and offers considerable gains on the Cityscapes, Pascal Context, and ADE20K datasets.

## 1 Introduction

Many important problems in computer vision, such as semantic segmentation and depth estimation, entail dense pixel-level predictions. Building accurate supervised models for these tasks is challenging because collecting ground truth annotations densely across all image pixels is costly, time-consuming, and error-prone. Accordingly, state-of-the-art techniques often resort to pretraining, where the model backbone (*i.e.,* encoder) is first trained as a supervised classifier (Sharif Razavian et al., 2014; Radford et al., 2021; Kolesnikov et al., 2020) or a self-supervised feature extractor (Oord et al., 2018; Hjelm et al., 2018; Bachman et al., 2019; He et al., 2020; Chen et al., 2020c;d; Grill et al., 2020). Backbone architectures such as ResNets (He et al., 2016) gradually reduce the feature map resolution. Hence, to conduct pixel-level prediction, a decoder is needed for upsampling back to the pixel level. Most state-of-the-art semantic segmentation models do not pre-train the additional parameters introduced by the decoder and initialize them at random. In this paper, we argue that random initialization of the decoder is far from optimal, and that pretraining the decoder weights with a simple but effective denoising approach can significantly improve performance.

---

[†]Work done as part of the Google AI Residency.
[*]Equal contribution.

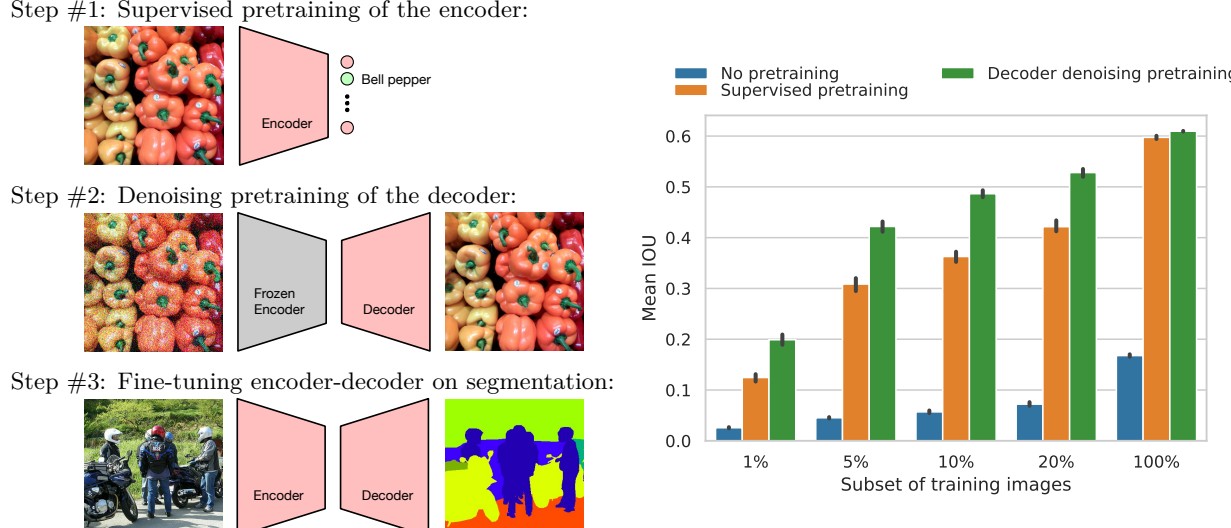

Figure 1: *Left:* An illustration of decoder denoising pretraining (DDeP). First, we train the encoder as a supervised classifier. Then, given a frozen encoder, we pretrain the decoder on the task of denoising. Finally the encoder-decoder model is fine-tuned on semantic segmentation. *Right:* Mean IoU on the Pascal Context dataset as a function of fraction of labeled training images available. Decoder denoising pretraining is particularly effective when a small number of labeled images is available, but continues to outperform supervised pretraining even on the full dataset. This demonstrates the importance of pretraining decoders for semantic segmentation, which was largely ignored in prior work.

Denoising autoencoders have a long and rich history in machine learning (Vincent et al., 2008; 2010). The general approach is to add noise to clean data and train the model to separate the noisy data back into clean data and noise components, which requires the model to learn the data distribution. Denoising objectives are well-suited for training dense prediction models because they can be defined easily on a per-pixel level. While the idea of denoising is old, denoising objectives have recently attracted new interest in the context of Diffusion Probabilistic Models (DPMs) for generative modeling (Sohl-Dickstein et al., 2015; Song & Ermon, 2019; Ho et al., 2020). Inspired by the renewed interest and success of denoising in diffusion models, we investigate the effectiveness of representations learned by denoising autoencoders for semantic segmentation, and in particular for pretraining decoder weights that are normally initialized randomly.

This paper studies pretraining of the decoders in semantic segmentation architectures and finds that significant gains can be obtained over random initialization, especially in the limited labeled data setting. We propose the use of denoising for decoder pretraining and connect denoising autoencoders to diffusion probabilistic models to improve various aspects of denoising pretraining such as the prediction of the noise instead of the image in the denoising objective and scaling of the image before adding Gaussian noise. This leads to a significant improvement over standard supervised pretraining of the encoder on three datasets. In Section 2, we give a brief overview before delving deeper into the details of generic denoising pretraining in Section 3 and decoder denosing pretraining in Section 4. Section 5 presents empirical comparisons with state-of-the-art methods.

## 2 Approach

Our goal is to learn image representations that can transfer well to dense visual prediction tasks. We consider an architecture comprising an encoder $f_\theta$ and a decoder $g_\phi$ parameterized by two sets of parameters $\theta$ and $\phi$. This model takes as input an image $\boldsymbol{x} \in \mathbb{R}^{H \times W \times C}$ and converts it into a dense representation $\boldsymbol{y} \in \mathbb{R}^{h \times w \times c}$, *e.g.,* a semantic segmentation mask.

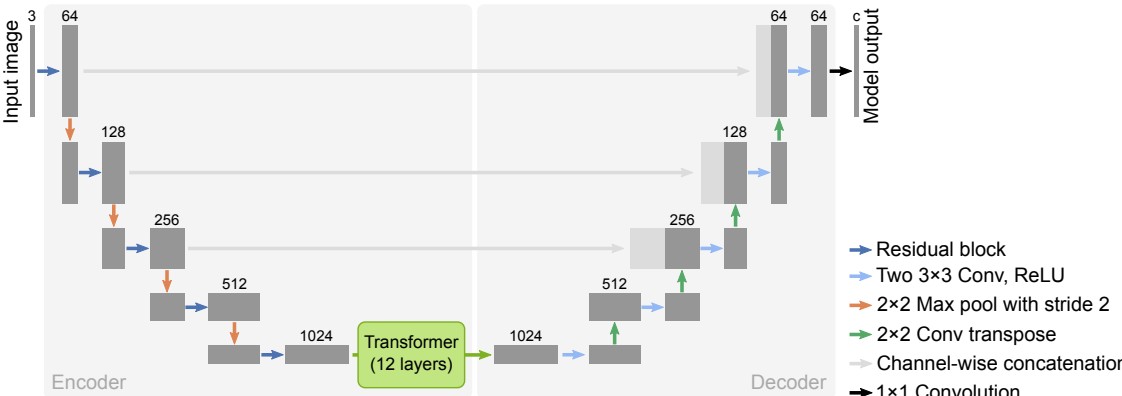

Figure 2: The Transformer-based UNet architecture used in our experiments. The encoder is a Hybrid-ViT model (Dosovitskiy et al., 2021).

We wish to find a way to initialize the parameters $\theta$ and $\phi$ such that the model can be effectively fine-tuned on semantic segmentation with a few labeled examples. For the encoder parameters $\theta$, we can follow standard practice and initialize them with weights pretrained on classification. Our main contribution concerns the decoder parameters $\phi$, which are typically initialized randomly. We propose to pretrain these parameters as a denoising autoencoder (Vincent et al., 2008; 2010): Given an unlabeled image $\boldsymbol{x}$, we obtain a noisy image $\widetilde{\boldsymbol{x}}$ by adding Gaussian noise $\sigma\boldsymbol{\epsilon}$ with a fixed standard deviation $\sigma$ to $\boldsymbol{x}$ and then train the model as an autoencoder $g_\phi \circ f_\theta$ to minimize the reconstruction error $\|g_\phi(f_\theta(\widetilde{\boldsymbol{x}})) - \boldsymbol{x}\|_2^2$ (optimizing only $\phi$ and holding $\theta$ fixed). We call this approach *Decoder Denoising Pretraining* (DDeP). Alternatively, both $\phi$ and $\theta$ can be trained by denoising (*Denoising Pretraining*; DeP). Below, we discuss several important modifications to the standard autoencoder formulation which we show to improve the quality of representations significantly.

As our experimental setup, we use a TransUNet (Chen et al. (2021); Figure 2). The encoder is pre-trained on ImageNet-21k (Deng et al., 2009) classification, while the decoder is pre-trained with our denoising approach, also using ImageNet-21k images without utilizing the labels. After pretraining, the model is fine-tuned on the Cityscapes, Pascal Context, or ADE20K semantic segmentation datasets (Cordts et al., 2016; Mottaghi et al., 2014; Zhou et al., 2018). We report the mean Intersection of Union (mIoU) averaged over all semantic categories. We describe further implementation details in Section 5.1.

Figure 1 shows that our DDeP approach significantly improves over encoder-only pretraining, especially in the few-shot regime. Figure 6 shows that even DeP, *i.e.,* denoising pretraining for the whole model (encoder and decoder) without any supervised pretraining, is competitive with supervised pretraining. Our results indicate that, despite its simplicity, denoising pretraining is a powerful method for learning semantic segmentation representations.

## 3 Denoising pretraining for both encoder and decoder

As introduced above, our goal is to learn effective visual representations that can transfer well to semantic segmentation and possibly other dense visual prediction tasks. We revisit denoising objectives to address this goal. We first introduce the standard denoising autoencoder formulation (for both encoder and decoder). We then propose several modifications of the standard formulation that are motivated by the recent success of diffusion models in image generation (Ho et al., 2020; Nichol & Dhariwal, 2021; Saharia et al., 2021b).

### 3.1 The standard denoising objective

In the standard denoising autoencoder formulation, given an unlabeled image $\boldsymbol{x}$, we obtain a noisy image $\widetilde{\boldsymbol{x}}$ by adding Gaussian noise $\sigma\boldsymbol{\epsilon}$ with a fixed standard deviation $\sigma$ to $\boldsymbol{x}$,

$$\widetilde{\boldsymbol{x}} = \boldsymbol{x} + \sigma\boldsymbol{\epsilon} , \qquad \boldsymbol{\epsilon} \sim \mathcal{N}(\boldsymbol{0}, \boldsymbol{I}) . \tag{1}$$

We then train an autoencoder $g_\phi \circ f_\theta$ to minimize the reconstruction error $\|g_\phi(f_\theta(\widetilde{\boldsymbol{x}})) - \boldsymbol{x}\|_2^2$. Accordingly, the objective function takes the form

$$O_{\mathrm{DeP}_1}(\theta, \phi \mid \sigma) = \mathbb{E}_{\boldsymbol{x}}\, \mathbb{E}_{\boldsymbol{\epsilon} \sim \mathcal{N}(\boldsymbol{0}, \boldsymbol{I})} \left\| g_\phi(f_\theta(\boldsymbol{x} + \sigma\boldsymbol{\epsilon})) - \boldsymbol{x} \right\|_2^2. \tag{2}$$

While this objective function already yields representations that are useful for semantic segmentation, we find that several key modifications can improve the quality of representations significantly.

## 3.2 Choice of denoising target in the objective

The standard denoising autoencoder objective trains a model to predict the noiseless image $\boldsymbol{x}$. However, diffusion models are typically trained to predict the noise vector $\boldsymbol{\epsilon}$ (Vincent, 2011; Ho et al., 2020):

$$O_{\mathrm{DeP}_2}(\theta, \phi \mid \sigma) = \mathbb{E}_{\boldsymbol{x}}\, \mathbb{E}_{\boldsymbol{\epsilon} \sim \mathcal{N}(\boldsymbol{0}, \boldsymbol{I})} \left\| g_\phi(f_\theta(\boldsymbol{x} + \sigma\boldsymbol{\epsilon})) - \boldsymbol{\epsilon} \right\|_2^2. \tag{3}$$

The two formulations would behave similarly for models with a skip connection from the input $\widetilde{\boldsymbol{x}}$ to the output. In that case, the model could easily combine its estimate of $\boldsymbol{\epsilon}$ with the input $\widetilde{\boldsymbol{x}}$ to obtain $\boldsymbol{x}$. However, in the absence of an explicit skip connection, our experiments show that predicting the noise vector is significantly better than predicting the noiseless image (Table 1).

| Method | full (2,975) | 1/4 (744) | 1/8 (372) | 1/30 (100) |
|---|---|---|---|---|
| Predict $x$ | 70.44 | 60.87 | 55.44 | 41.40 |
| Predict $\epsilon$ | **75.01** | **67.26** | **61.94** | **48.36** |

Table 1: Comparison of noise prediction and image prediction on Cityscapes.

## 3.3 Scalability of denoising as a pretraining objective

Unsupervised pretraining methods are ultimately limited by the mismatch between the representations learned by the pretraining objective and the representations needed for the final target task. An important "sanity check" for any unsupervised objective is that it does not reach this limit quickly, to ensure that it is well-aligned with the target task. We find that representations learned by denoising continue to improve up to the our maximal feasible pretraining compute budget (Figure 3). This suggests that denoising is a scalable approach and that the representation quality will continue to improve as compute budgets increase.

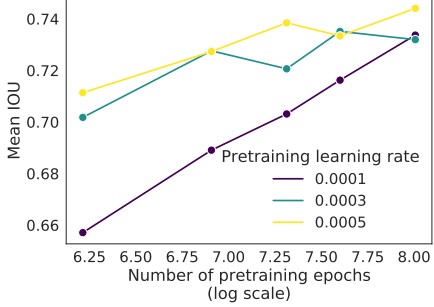

Figure 3: Effect of length of pretraining duration on downstream performance. Cityscapes is used for pretraining and downstream finetuning.

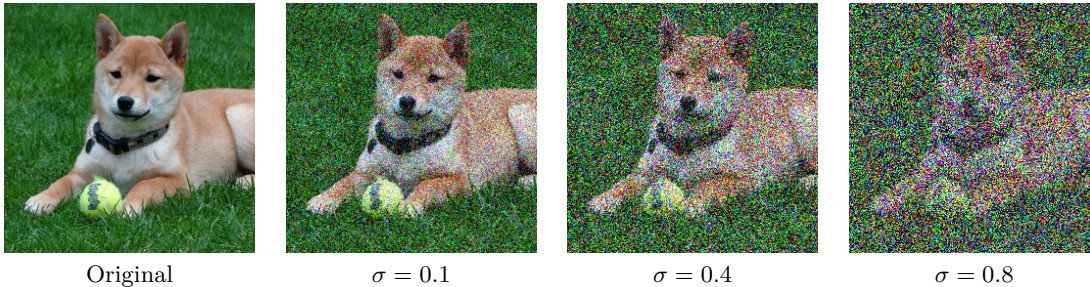

| Original | $\sigma = 0.1$ | $\sigma = 0.4$ | $\sigma = 0.8$ |

Figure 4: An illustration of a $256 \times 256$ image and a few reasonable values of standard deviation ($\sigma$) for Gaussian noise. For visualization, we clip noisy pixel values to $[0, 1]$, but during training no clipping is used.

### 3.4 Denoising versus supervised pretraining

In the standard denoising autoencoder formulation, the whole model (encoder and decoder) is trained using denoising. However, denoising pretraining of the full model underperforms standard supervised pretraining of the encoder, at least when fine-tuning data is abundant (Table 2). In the next section, we explore combining denoising and supervised pretraining to obtain the benefits of both.

| Method | 100% (2,975) | 25% (744) | 2% (60) | 1% (30) |
|---|---|---|---|---|
| No Pretraining | 63.47 | 39.63 | 21.23 | 18.52 |
| Supervised | **80.36** | **75.55** | 41.33 | 35.51 |
| Denoising Pretraining | 77.14 | 68.87 | **42.79** | **37.55** |

Table 2: Performance of Denoising Pretraining on the Cityscapes validation set. *No Pretraining* refers to random initialization of the model; *Supervised* refers to ImageNet-21k classification pretraining of the encoder and random initialization of the decoder; *Denoising Pretraining* refers to unsupervised denoising pretraining of the whole model. The Denoising model is pretrained in an unsupervised fashion for 6000 epochs using all Cityscapes images, with a noise magnitude of $\sigma = 0.8$. Denoising Pretraining performs strongly in the limited labeled data regime, but falls behind supervised pretraining when labeled data is abundant.

## 4 Denoising pretraining for decoder only

In practice, since strong and scalable methods for pretraining the encoder weights already exist, the main potential of denoising lies in pretraining the decoder weights. To this end, we fix the encoder parameters $\theta$ at the values obtained through supervised pretraining on ImageNet-21k, and pretrain only the decoder parameters $\phi$ with denoising, leading to the following objective:

$$O_{\mathrm{DDeP}_1}(\phi \mid \theta, \sigma) = \mathbb{E}_{\boldsymbol{x}} \, \mathbb{E}_{\boldsymbol{\epsilon} \sim \mathcal{N}(\boldsymbol{0}, \boldsymbol{I})} \left\| g_\phi(f_\theta(\boldsymbol{x} + \sigma \boldsymbol{\epsilon})) - \boldsymbol{\epsilon} \right\|_2^2 . \tag{4}$$

We refer to this pretraining scheme as Decoder Denoising Pretraining (DDeP). As we show below, DDeP performs better than either pure supervised or pure denoising pretraining across all label-efficiency regimes. We investigate the key design decisions of DDeP such as the noise formulation and the optimal noise level in this section before presenting benchmark results in Section 5.

### 4.1 Noise magnitude and relative scaling of image and noise

The key hyperparameter for decoder denoising pretraining is the magnitude of noise that is added to the image. The noise variance $\sigma$ must be large enough that the network learns meaningful image representations

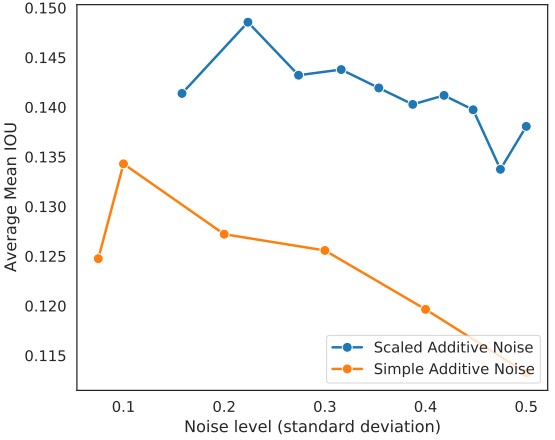

| Segmentation dataset | Noise type | 100% | 10% | 5% | 1% |
|---|---|---|---|---|---|
| Pascal Context | Simple | 59.64 | 44.70 | 39.14 | 18.05 |
| Pascal Context | Scaled | **60.00** | **48.27** | **41.93** | **19.69** |
| ADE20K | Simple | 48.88 | 34.21 | 24.07 | 9.17 |
| ADE20K | Scaled | **48.97** | **34.99** | **26.86** | **10.80** |

Figure 5: *Left:* Effect of noise magnitude on downstream performance. Results are on 1% of labelled examples and averaged over Pascal Context and ADE20K. *Right:* Performance comparison of different noise formulations. Scaled additive noise formulation consistently outperforms the simple additive noise formulation.

in order to remove it, but not so large that it causes excessive distribution shift between clean and noisy images. For visual inspection, Figure 4 illustrates a few example values of $\sigma$.

In addition to the absolute magnitude of the noise, we find that the relative scaling of clean and noisy images also plays an important role. Different denoising approaches differ in this aspect. Specifically, DDPMs generate a noisy image $\widetilde{\boldsymbol{x}}$ as

$$\widetilde{\boldsymbol{x}} \; = \; \sqrt{\gamma}\,\boldsymbol{x} + \sqrt{1-\gamma}\,\boldsymbol{\epsilon} \; = \; \frac{1}{\sqrt{1+\sigma^2}}\,(\boldsymbol{x} + \sigma\,\boldsymbol{\epsilon}) \qquad \boldsymbol{\epsilon} \sim \mathcal{N}(\mathbf{0}, \boldsymbol{I}) \; . \tag{5}$$

This differs from the standard denoising formulation in Eq. (1) in that $\boldsymbol{x}$ is attenuated by $\sqrt{\gamma}$ and $\boldsymbol{\epsilon}$ is attenuated by $\sqrt{1-\gamma}$ to ensure that the variance of the random variables $\widetilde{\boldsymbol{x}}$ is 1 if the variance of $\boldsymbol{x}$ is 1. With this formulation, our denoising pretraining objective becomes:

$$O_{\mathrm{DDeP_2}}(\phi \mid \theta, \sigma) = \mathbb{E}_{\boldsymbol{x}}\,\mathbb{E}_{\boldsymbol{\epsilon} \sim \mathcal{N}(\mathbf{0}, \boldsymbol{I})} \left\| g_\phi(f_\theta(\frac{1}{\sqrt{1+\sigma^2}}\,(\boldsymbol{x} + \sigma\,\boldsymbol{\epsilon}))) - \boldsymbol{\epsilon} \right\|_2^2 \; . \tag{6}$$

In Figure 5, we compare this *scaled additive noise* formulation with the *simple additive noise* formulation (Eq. (1)) and find that scaling the images delivers notable gains in downstream semantic segmentation performance. We speculate that the decoupling of the variance of the noisy image from the noise magnitude reduces the distribution shift between clean and noisy images, which improves transfer of the pre-trained representations to the final task. Hence this formulation is used for the rest of the paper. We find the optimal noise magnitude to be 0.22 (Figure 5) for the scaled additive noise formulation and use that value for the experiments below.

## 4.2 Choice of pretraining dataset

In principle, any image dataset can be used for denoising pretraining. Ideally, we would like to use a large dataset such as ImageNet for pretraining, but this raises the potential concern that distribution shift between pretraining and target data may impact performance on the target task. To test this, we compare Decoder Denoising Pretraining on a few datasets while the encoder is pretrained on ImageNet-21K with classification objective and kept fixed. We find that pretraining the decoder on ImageNet-21K leads to better results than pretraining it on the target data for all tested datasets (Cityscapes, Pascal Context, and ADE20K; Table 3). Notably, this holds even for Cityscapes, which differs significantly in terms of image distribution

from ImageNet-21k. Models pretrained with DDeP on a generic image dataset are therefore generally useful across a wide range of target datasets.

| Segmentation dataset | Decoder pretraining dataset | 100% | 10% | 5% |
|---|---|---|---|---|
| Pascal Context | Pascal VOC | 60.13 | 49.95 | 44.30 |
| Pascal Context | ImageNet-21K | **60.57** | **50.61** | **45.13** |
| ADE20K | ADE20K | 48.92 | 36.14 | 28.49 |
| ADE20K | ImageNet-21K | **49.37** | **37.14** | **29.74** |
| Cityscapes (fine) | Cityscapes (fine & coarse) | 80.53 | 72.67 | 62.23 |
| Cityscapes (fine) | ImageNet-21K | **80.72** | **73.21** | **66.51** |

Table 3: Ablation of the dataset used for decoder denoising pretraining. ImageNet-21K pretraining performs better than target dataset pretraining in all the settings.

### 4.3 Decoder variants

Given that decoder denoising pretraining significantly improves over random initialization of the decoder, we hypothesized that the method could allow scaling up the size of the decoder beyond the point where benefits diminish when using random initialization. We test this by varying the number of feature maps at the various stages of the decoder. The default $(1\times)$ decoder configuration for all our experiments is $[1024, 512, 256, 128, 64]$ where the value at index $i$ corresponds to the number of feature maps at the $i^{th}$ decoder block. This is reflected in Figure 2. On Cityscapes, we experiment with doubling the default width of all decoder layers $(2\times)$, while on Pascal Context and ADE20K, we experiment with tripling $(3\times)$ the widths. While larger decoders usually improve performance even when initialized randomly, DDeP leads to additional gains in all cases. DDeP may therefore unlock new decoder-heavy architectures. We present our main results in Section 5 for both $1\times$ decoders and $2\times/3\times$ decoders.

### 4.4 Extension to diffusion process

Above, we find that pre-trained representations can be improved by adapting some aspects of the standard autoencoder formulation, such as the choice of the prediction target and the relative scaling of image and noise, to be more similar to diffusion models. This raises the question whether representations could be further improved by using a full diffusion process for pretraining. Here, we study extensions that bring the method closer to the full diffusion process used in DDPMs, but find that they do not improve results over the simple method discussed above.

**Variable noise schedule.** Since it uses a single fixed noise level ($\gamma$ in Eq. (5)), our method corresponds to a single step in a diffusion process. Full DDPMs model a complete diffusion process from a clean image to pure noise (and its reverse) by sampling the noise magnitude $\gamma$ uniformly at random from $[0, 1]$ for each training example (Ho et al., 2020). We therefore also experiment with sampling $\gamma$ randomly, but find that a fixed $\gamma$ performs best (Table 4).

**Conditioning on noise level.** In the diffusion formalism, the model represents the (reverse) transition function from one noise level to the next, and is therefore conditioned on the current noise level. In practice, this is achieved by supplying the $\gamma$ sampled for each training example as an additional model input, e.g. to normalization layers. Since we typically use a fixed noise level, conditioning is not required for our method. Conditioning also provides no improvements when using a variable noise schedule.

**Weighting of noise levels.** In DDPMs, the relative weighting of different noise levels in the loss has a large impact on sample quality (Ho et al., 2020). Since our experiments suggest that multiple noise levels are not necessary for learning transferable representations, we did not experiment with the weighting of different noise levels, but note that this may be an interesting direction for future research.

| Method | 100% (4,998) | 20% (1,000) | 10% (500) |
|---|---|---|---|
| DDeP $\gamma \sim U(0.9, 0.95)$ | 59.71 | 52.53 | 49.23 |
| DDeP $\gamma = 0.95$ | **59.97** | **53.36** | **49.84** |

Table 4: Comparison of fixed value of $\sigma$ with uniform sampling of $\sigma$ in the interval $[0.9, 0.95]$ on Pascal Context, with a $3\times$ width decoder. Labeled examples are varied from 100% to 10% of the original TRAIN set, and mIoU (%) on the VALIDATION set is reported.

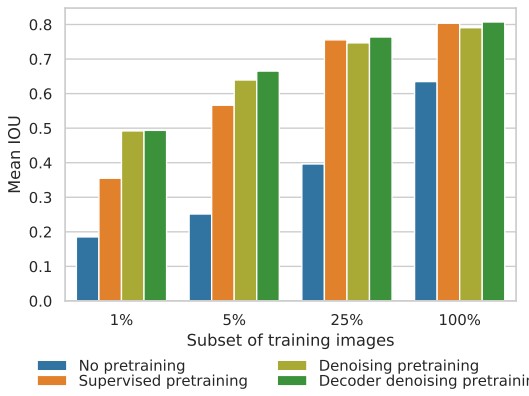

| Method | Decoder width | full (2,975) | 1/4 (744) | 1/8 (372) | 1/30 (100) |
|---|---|---|---|---|---|
| No Pretraining | $1\times$ | 63.47 | 39.63 | 34.74 | 25.79 |
| Supervised | $1\times$ | 80.36 | 75.55 | 72.56 | 54.72 |
| DeP | $1\times$ | 79.07 | 74.68 | 70.36 | 61.79 |
| DDeP | $1\times$ | **80.72** | **76.38** | **73.21** | **64.48** |
| No Pretraining | $2\times$ | 62.25 | 37.72 | 33.73 | 24.93 |
| Supervised | $2\times$ | 80.50 | 75.57 | 72.84 | 60.36 |
| DDeP | $2\times$ | **80.91** | **76.86** | **73.81** | **64.75** |

Figure 6: Cityscapes mIoU on VAL_FINE set. Labeled examples are varied from full to 1/30 of the original TRAIN_FINE set *M*ean IoU on the Cityscapes validation set as a function of fraction of labeled training images available. Denoising pretraining is particularly effective when less than 5% of labeled images is available. Supervised pretraining of the backbone on ImageNet-21K outperforms denoising pretraining when label fraction is larger. Decoder denoising pretraining offers the best of both worlds, achieving competitive results across label fractions.

# 5 Benchmark results

We assess the effectiveness of the proposed Decoder Denoising Pretraining (**DDeP**) on several semantic segmentation datasets and conduct label-efficiency experiments.

## 5.1 Implementation details

For downstream fine-tuning of the pretrained models for the semantic segmentation task, we use the standard pixel-wise cross-entropy loss. We use the Adam (Kingma & Ba, 2015) optimizer with a cosine learning rate decay schedule. For Decoder Denoising Pretraining (DDeP), we use a batch size of 512 and train for 100 epochs. The learning rate is $6e-5$ for the $1\times$ and $3\times$ width decoders, and $1e-4$ for the $2\times$ width decoder.

When fine-tuning the pretrained models on the target semantic segmentation task, we sweep over weight decay and learning rate values between $[1e-5, 3e-4]$ and choose the best combination for each task. For the 100% setting, we report the means of 10 runs on all of the datasets. On Pascal Context and ADE20K, we also report the mean of 10 runs (with different subsets) for the 1%, 5% and 10% label fractions and 5 runs for the 20% setting. On Cityscapes, we report the mean of 10 runs for the 1/30 setting, 6 runs for the 1/8 setting and 4 runs for the 1/4 setting.

During training, random cropping and random left-right flipping is applied to the images and their corresponding segmentation masks. We randomly crop the images to a fixed size of $1024 \times 1024$ for Cityscapes and $512 \times 512$ for ADE20K and Pascal Context. All of the decoder denoising pretraining runs are conducted at a $224 \times 224$ resolution.

| Method | full (2,975) | 1/4 (744) | 1/8 (372) | 1/30 (100) |
|---|---|---|---|---|
| AdvSemSeg (Hung et al., 2018) | - | 62.3 | 58.8 | - |
| s4GAN (Mittal et al., 2021) | 65.8 | 61.9 | 59.3 | - |
| DMT (Feng et al., 2020b) | 68.16 | - | 63.03 | 54.80 |
| ClassMix (Olsson et al., 2021) | - | 63.63 | 61.35 | - |
| CutMix (French et al., 2019) | - | 68.33 | 65.82 | 55.71 |
| PseudoSeg (Zou et al., 2021) | - | 72.36 | 69.81 | 60.96 |
| Sup. baseline (Zhong et al., 2021) | 74.88 | 73.31 | 68.72 | 56.09 |
| PC$^2$Seg (Zhong et al., 2021) | 75.99 | 75.15 | 72.29 | 62.89 |
| DDeP (Ours) | **80.91** | **76.86** | **73.81** | **64.75** |

Table 5: Comparison with the state-of-the-art on Cityscapes. The result of French et al. (2019) is reproduced by Zou et al. (2021) based on DeepLab-v3+, while the results of Hung et al. (2018); Mittal et al. (2021); Feng et al. (2020b); Olsson et al. (2021) are based on DeepLab-v2. All of the baselines except ours make use of a ResNet-101 backbone, and we emphasize that this comparison confounds differences in architecture and pretraining strategy.

During inference on Cityscapes, we evaluate on the full resolution $1024 \times 2048$ images by splitting them into two $1024 \times 1024$ input patches. We apply horizontal flip and average the results for each half. The two halves are concatenated to produce the full resolution output. For Pascal Context and ADE20K, we also use multi-scale evaluation with rescaled versions of the image in addition to the horizontal flips. The scaling factors used are (0.5, 0.75, 1.0, 1.25, 1.5, 1.75).

## 5.2 Performance gain by decoder denoising pretraining

On Cityscapes, DDeP outperforms both DeP and supervised pretraining. In Figure 6, we report the results of DeP and DDeP on Cityscapes and compare them with the results of training from random initialization or initializing with an ImageNet-21K-pretrained encoder. The DeP results make use of the scaled additive noise formulation (Equation (5)) leading to a significant boost in performance over the results obtained with the standard denoising objective. As shown in Figure 6, DeP outperforms the supervised baseline in the 1% and 5% labelled images settings. Decoder Denoising Pretraining (DDeP) further improves over both DeP and ImageNet-21K supervised pretraining for both the 1× and 2× decoder variants (Table Figure 6).

DDeP outperforms previously proposed methods for label-efficient semantic segmentation on Cityscapes at all label fractions, as shown in Table 5.2. With only 25% of the training data, DDeP produces better segmentations than the strongest baseline method, PC$^2$Seg (Zhong et al., 2021), does when trained on the full dataset. Unlike most recent work, we do not perform evaluation at multiple scales on Cityscapes, which should lead to further improvements.

DDeP also improves over supervised pretraining on the Pascal Context dataset. Figure 1 compares the performance of DDeP with that of the supervised baseline and a randomly initialized model on Pascal Context on $1\%, 5\%, 10\%, 20\%$ and $100\%$ of the training data. Table 5.2 compares these results with those obtained with a 3× decoder. For both 1× and 3× decoders, DDeP significantly outperforms architecturally identical supervised models, obtaining improvements of 4-12% mIOU across all semi-supervised settings. Notably, with only 10% of the labels, DDeP outperforms the supervised model trained with 20% of the labels.

Figure 7 shows similar improvements from DDeP on the ADE20K dataset. Again, we see gains of more than 10 points in the 5% and 10% settings and 5 points in the 1% setting. These consistent results demonstrate the effectiveness of DDeP across datasets and training set sizes.

Our results above use a TransUNet (Chen et al. (2021); Figure 2) architecture to attain maximal performance, but DDeP is backbone-agnostic and provides gains when used with simpler backbone architectures as well. In Table 7, we train a standard U-Net with a ResNet50 encoder with DDeP on Pascal Context (without

| Method | Decoder width | 100% (4,998) | 20% (1,000) | 10% (500) | 5% (250) | 1% (50) |
|---|---|---|---|---|---|---|
| No pretraining | 1× | 16.78 | 7.21 | 5.69 | 4.53 | 2.57 |
| Supervised | 1× | 59.74 | 42.15 | 36.28 | 30.86 | 12.45 |
| DDeP | 1× | **60.95** | **52.81** | **48.64** | **42.20** | **19.90** |
| No pretraining | 3× | 17.22 | 7.32 | 6.16 | 4.97 | 2.95 |
| Supervised | 3× | 61.25 | 51.49 | 44.71 | 37.52 | 12.53 |
| DDeP | 3× | **62.04** | **55.28** | **51.55** | **46.29** | **24.69** |

Table 6: Pascal Context mIoU (%) on the VALIDATION set for labeled examples varied from 100% to 1% of the original TRAIN set. Supervised indicates ImageNet-21K pretraining of the backbone

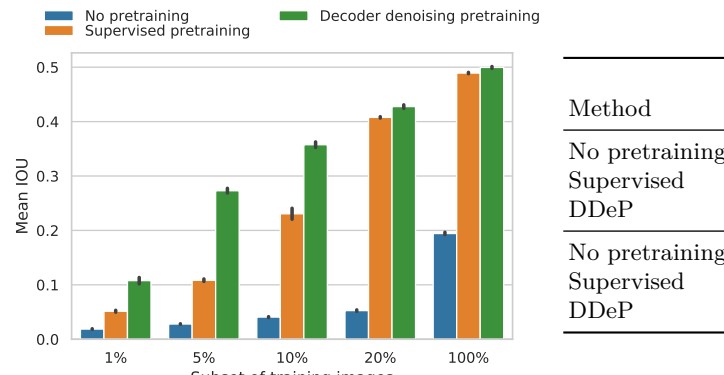

| Method | Decoder width | 100% (20,210) | 20% (4,042) | 10% (2,021) | 5% (1,010) | 1% (202) |
|---|---|---|---|---|---|---|
| No pretraining | 1× | 19.43 | 5.26 | 4.07 | 2.80 | 1.87 |
| Supervised | 1× | 48.92 | 40.77 | 23.05 | 10.84 | 5.14 |
| DDeP | 1× | **49.96** | **42.76** | **35.75** | **27.29** | **10.77** |
| No pretraining | 3× | 16.67 | 5.88 | 4.20 | 2.91 | 1.90 |
| Supervised | 3× | 49.60 | 41.65 | 33.04 | 16.40 | 5.31 |
| DDeP | 3× | **50.88** | **43.26** | **39.01** | **32.30** | **16.30** |

Figure 7: ADE20K mIoU (%) on the VALIDATION set for labeled examples varied from 100% to 1% of the original TRAIN set. Supervised indicates ImageNet-21K pretraining of the backbone.

multi-scale evaluation). DDeP outperforms the supervised baseline in all settings showing that our method generalizes beyond transformer architectures.

| Method | Decoder wd. | 100% | 20% | 10% | 5% | 1% |
|---|---|---|---|---|---|---|
| No pretraining | 1× | 19.01 | 8.46 | 6.72 | 5.30 | 2.73 |
| Supervised | 1× | 45.21 | 24.55 | 19.27 | 14.97 | 6.09 |
| DDeP | 1× | **46.07** | **30.38** | **26.39** | **21.12** | **9.63** |

Table 7: Performance of a U-Net with a simple ResNet50 backbone on Pascal Context.

# 6 Related work

Because collecting detailed pixel-level labels for semantic segmentation is costly, time-consuming, and error-prone, many methods have been proposed to enable semantic segmentation from fewer labeled examples (Tarvainen & Valpola, 2017; Miyato et al., 2018; Hung et al., 2018; Mittal et al., 2021; French et al., 2019; Ouali et al., 2020; Zou et al., 2021; Feng et al., 2020b; Ke et al., 2020; Olsson et al., 2021; Zhong et al., 2021). These methods often resort to semi-supervised learning (SSL) (Chapelle et al., 2006; Van Engelen & Hoos, 2020), in which one assumes access to a large dataset of unlabeled images in addition to labeled training data. In what follows, we will discuss previous work on the role of strong data augmentation, generative models, self-training, and self-supervised learning in label-efficient semantic segmentation. While this work focuses on self-supervised pretraining, we believe strong data augmentation and self-training can be combined with the proposed denoising pretraining approach to improve the results even further.

**Data augmentation.** French *et al.* (French et al., 2019) demonstrate that strong data augmentation techniques such as Cutout (DeVries & Taylor, 2017) and CutMix (Yun et al., 2019) are particularly effective

for semantic segmentation from few labeled examples. Ghiasi et al. (2021) find that a simple copy-paste augmentation is helpful for instance segmentation. Previous work (Remez et al., 2018; Chen et al., 2019; Bielski & Favaro, 2019; Arandjelović & Zisserman, 2019) also explores completely unsupervised semantic segmentation by leveraging GANs (Goodfellow et al., 2014) to compose different foreground and background regions to generate new plausible images. We make use of relatively simple data augmentation including horizontal flip and random inception-style crop (Szegedy et al., 2015). Using stronger data augmentation is left to future work.

**Generative models.** Early work on label-efficient semantic segmentation uses GANs to generate synthetic training data (Souly et al., 2017) and to discriminate between real and predicted segmentation masks (Hung et al., 2018; Mittal et al., 2021). DatasetGAN (Zhang et al., 2021) shows that modern GAN architectures (Karras et al., 2019) are effective in generating synthetic data to help pixel-level image understanding, when only a handful of labeled images are available. Our method is highly related to Diffusion and score-based generative models (Sohl-Dickstein et al., 2015; Song & Ermon, 2019; Ho et al., 2020), which represent an emerging family of generative models resulting in image sample quality superior to GANs (Dhariwal & Nichol, 2021; Ho et al., 2021). These models are linked to denoising autoencoders through denoising score matching (Vincent, 2011) and can be seen as methods to train energy-based models (Hyvärinen & Dayan, 2005). Denoising Diffusion Models (DDPMs) have recently been applied to conditional generation tasks such as super-resolution, colorization, and inpainting (Li et al., 2021; Saharia et al., 2021b; Song et al., 2021; Saharia et al., 2021a), suggesting these models may be able to learn useful image representations. We are inspired by the success of DDPMs, but we find that many components of DDPMs are not necessary and simple denoising pretraining works well. Diffusion models have been used to iteratively refine semantic segmentation masks (Amit et al., 2021; Hoogeboom et al., 2021). Baranchuk et al. (Baranchuk et al., 2021) demonstrates the effectiveness of features learned by diffusion models for semantic segmentation from very few labeled examples. By contrast, we utilize simple denoising pretraining for representation learning and study full fine-tuning of the encoder-decoder architecture as opposed to extracting fixed features. Further, we use well-established benchmarks to compare our results with prior work.

**Self-training, consistency regularization.** *Self-training* (self-learning or pseudo-labeling) is one of the oldest SSL algorithms (Scudder, 1965; Fralick, 1967; Agrawala, 1970; Yarowsky, 1995). It works by using an initial supervised model to annotate unlabeled data with so-called *pseudo labels*, and then uses a mixture of pseudo- and human-labeled data to train improved models. This iterative process may be repeated multiple times. Self-training has been used to improve object detection (Rosenberg et al., 2005; Zoph et al., 2020) and semantic segmentation (Zhu et al., 2020; Zou et al., 2021; Feng et al., 2020a; Chen et al., 2020a). Consistency regularization is closely related to self-training and enforces consistency of predictions across augmentations of an image (French et al., 2019; Kim et al., 2020; Ouali et al., 2020). These methods often require careful hyper-parameter tuning and a reasonable initial model to avoid propagating noise. Combining self-training with denosing pretraining will likely improve the results further.

**Self-supervised learning.** Self-supervised learning methods formulate predictive pretext tasks that are easy to construct from unlabeled data and can benefit downstream discriminative tasks. In natural language processing (NLP), the task of masked language modeling (Devlin et al., 2019; Liu et al., 2019; Raffel et al., 2020) has become the de facto standard, showing impressive results across NLP tasks. In computer vision, different pretext tasks for self-supervised learning have been proposed, including the task of predicting the relative positions of neighboring patches within an image (Doersch et al., 2015), the task of inpainting (Pathak et al., 2016), solving Jigsaw Puzzles (Noroozi & Favaro, 2016), image colorization (Zhang et al., 2016; Larsson et al., 2016), rotation prediction (Gidaris et al., 2018), and other tasks (Zhang et al., 2017; Caron et al., 2018; Kolesnikov et al., 2019). Recently, methods based on exemplar discrimination and contrastive learning have shown promising results for image classification (Oord et al., 2018; Hjelm et al., 2018; He et al., 2020; Chen et al., 2020c;d; Grill et al., 2020). These approaches have been used to successfully pretrain backbones for object detection and segmentation (He et al., 2020; Chen et al., 2020e), but unlike this work, they typically initialize decoder parameters at random. Recently, there are also a family of emerging methods based on masked auto-encoding, such as BEIT (Bao et al., 2021), MAE (He et al., 2021), and others (Zhou et al., 2021; Dong et al., 2021; Chen et al., 2022). We note that our approach is developed concurrently to this

family of mask image modeling, and our technique is also orthogonal in that we focus on decoder pretraining, which was not the focus of aforementioned papers.

**Self-supervised learning for dense prediction.** Pinheiro et al. (2020) and Wang et al. (2021) propose dense contrastive learning, an approach to self-supervised pretraining for dense prediction tasks, in which contrastive learning is applied to patch- and pixel-level features as opposed to image level-features. This is reminiscent of AMDIM (Bachman et al., 2019) and CPC V2 (Hénaff et al., 2019). Zhong et al. (2021) take this idea further and combine segmentation mask consistency between the output of the model for different augmentations of an image (possibly unlabeled) and pixel-level feature consistency across augmentations.

**Transformers for vision.** Inspired by the success of Transformers in NLP (Vaswani et al., 2017), several publications study combining convolution and self-attention for object detection (Carion et al., 2020), semantic segmentation (Wang et al., 2018; 2020b), and panoptic segmentation (Wang et al., 2020a). Vision Transformer (ViT) (Dosovitskiy et al., 2021) demonstrates that a convolution-free approach can yield impressive results when a massive labeled dataset is available. Recent work has explored the use of ViT as a backbone for semantic segmentation (Zheng et al., 2020; Liu et al., 2021; Strudel et al., 2021). These approaches differ in the structure of the decoder, but they show the power of ViT-based semantic segmentation. We adopt a hybrid ViT (Dosovitskiy et al., 2021) as the backbone, where the patch embedding projection is applied to patches extracted from a convolutional feature map. We study the size of the decoder, and find that wider decoders often improve semantic segmentation results.

## 7    Discussion

The second step of decoder denoising pretraining adds a computational overhead which makes DDeP costlier to train than supervised pretraining of the encoder. Indeed, training DDeP costs 117.6 PFLOPs compared to 48.3 PFLOPs for the supervised baseline on 32 TPU-v4 chips. This is a substantial amount of additional compute. However, DDeP only has to be performed once, for all downstream uses since our results in Table 3 show that ImageNet-21K pretraining transfers well to several downstream tasks. Hence, the pretraining cost is paid only once and is then amortized over all downstream uses of the checkpoint.

In this work, we only explored initializing the encoder from a supervised model it is the most established encoder initialization scheme that provides state-of-the-art performance. However, other encoder initialization schemes like SimCLR (Chen et al., 2020b) may provide interesting alternatives.

Given the results obtained with denoising, it is interesting to explore other self-supervised learning approaches like image colourization and inpainting for the purposes of decoder pretraining. However, preliminary experiments with these other approaches didn't yield better performance than denoising. It is also interesting to explore the use of denoising pretraining for other dense prediction tasks.

## 8    Conclusion

Inspired by the recent popularity of diffusion probabilistic models for image synthesis, we investigate the effectiveness of these models in learning useful transferable representations for semantic segmentation. Surprisingly, we find that pretraining a semantic segmentation model as a denoising autoencoder leads to large gains in semantic segmentation performance, especially when the number of labeled examples is limited. We build on this observation and propose a two-stage pretraining approach in which supervised pretrained encoders are combined with denoising pretrained decoders. This leads to consistent gains across datasets and training set sizes, resulting in a practical approach to pretraining.

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

# A   Appendix

## A.1   Decoder denoising pretraining on an ImageNet-1K backbone

In Tab. 8 we perform decoder denoising pretraining on an ImageNet-1K backbone and compare with supervised pretraining on the Cityscapes dataset. In this setting, DDeP outperforms the supervised training baseline which also uses an ImageNet-1k pretrained backbone. This shows that the gains of DDeP is not dependent on the specific dataset used for the pretraining of the backbone.

| Method | Decoder width | full (2,975) | 1/4 (744) | 1/8 (372) | 1/30 (100) |
|---|---|---|---|---|---|
| No Pretraining | $1\times$ | 63.47 | 39.63 | 34.74 | 25.79 |
| Supervised | $1\times$ | 77.11 | 70.04 | 65.07 | 52.97 |
| DDeP | $1\times$ | **77.86** | **70.77** | **66.01** | **54.11** |

Table 8: Performance of decoder denoising pretraining on an ImageNet-1K backbone.

## A.2   Visualization of decoder denoising pretraining

In Fig. 8, we visualize the output of the decoder denoising pretraining step which shows that our decoder-only denoising pretraining approach is able to rightly separate image and noise.

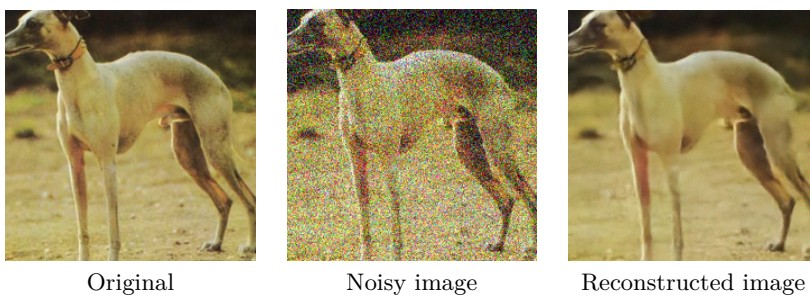

Original      Noisy image      Reconstructed image

Figure 8: Visualization of the output decoder denoising pretraining step.

