# OpenReview forum: "Decoder Denoising Pretraining for Semantic Segmentation"
_TMLR — Accepted by TMLR_

### Review · Reviewer_B1kf · 2022-05-29

**Summary Of Contributions:**



(Summury) In this paper, the authors propose a self-supervised pre-training method focusing on decoders. The proposed method is simple, effective, and inspiring. The authors made a comprehensive analysis and made many fascinating observations. I absolutely love this article and think it opens a great door for the community. I think this article is very suitable for publication in TMLR. Although the authors' method may have certain unfairness when compared with competitors.



**Broader Impact Concerns:**


I do not have a concern about the ethical implications of this work.



**Requested Changes:**


Please refer to the negative reviews in "*Strengths And Weaknesses."


**Strengths And Weaknesses:**


(Positive) The authors propose a pre-training scheme for the decoder, which I think is extremely reasonable. Since the encoder is already pre-trained at the semantic level, there is no need to pre-train with self-supervision based on noise prediction. The author only focuses on the decoder part; the idea is wonderful. This article is very meaningful. The results in Figure 6 also confirm that pre-training the encoder with a generative model is worse than pre-training with classification and then fixing it without learning.

(Positive) Most of the existing papers do ignore the pre-training of the decoder and only use randomization. The author attaches great importance to this issue, which is very admirable.

(Negative) However, it is unreasonable for the author to associate this article with DDPM. As a generative model, DDPM's ability to learn discriminative representations is controversial. Intuitively, using DDPM to replace the authors' approach is complicated and ineffective. This is also verified by the authors' experiments. Authors do not need to force a relationship with DDPM in the introduction chapter. The authors' articles are good enough that they don't need to take advantage of the hot spots of DDPM.

(Negative) It can be seen from Figure 2 and the multiple experimental results of the article that when the sample size of the downstream task increases, the advantages of the authors' method decrease. Please give an explanation.

(Negative) I have an explanation that does not favor the authors. Using pre-training is equivalent to optimizing for a longer time. According to "rethinking ImageNet pre-training," on downstream tasks, if the training is long enough, even without pre-training, good results can be achieved. Therefore, all the experimental settings of the authors may be unfair. Perhaps experiments of training for longer on downstream tasks are needed.

(Positive) I appreciate that the authors have a discussion and experimental analysis on whether it is better to predict the original image or predict the noise. This is a great experiment and provides readers with great insight.


(Positive) For the authors' analysis of "Scalability of denoising as a pretraining objective" in Section 3.3, I think it makes perfect sense. This should be a model for the community.


(Negative) Is the following statement in Section 3.4 wrong?
"However, denoising pretraining of the full model underperforms standard supervised pretraining of the encoder, at least when fine-tuning data is abundant."


(Positive) The following observations by the authors are very insightful, and I appreciate them very much.

"We find that scaling the images delivers notable gains in downstream semantic segmentation performance."

The following explanations by the authors are also very reasonable, and I agree with them.

"We speculate that the decoupling of the variance of the noisy image from the noise magnitude reduces the distribution shift between clean and noisy images, which improves transfer of the pre-trained representations to the final task."


(Positive) The authors' following observations are sound and deserve the attention of the community. I think it's significant.

"We find that pretraining the decoder on ImageNet-21K leads to better results than pretraining it on the target data for all tested datasets (Cityscapes, Pascal Context, and ADE20K; Table 3). Notably, this holds even for Cityscapes, which differs significantly in terms of image distribution from ImageNet-21k. Models pretrained with DDeP on a generic image dataset are therefore generally useful across a wide range of target datasets."


(Positive) The authors study the performance of decoders with different depths under the pre-training method proposed by the authors. The result is good. Its explanation is reasonable.

---

> ### Author Response · Authors · 2022-07-07
> **Response to Reviewer B1kf.**
>
> Thank you for your valuable feedback! We address your questions below, focusing on those marked as negative.
>
> >”It is unreasonable for the author to associate this article with DDPM.”
>
> We agree that unnecessary “hype” should be avoided and will reduce the emphasis on DDPMs in the introduction.
>
> However, some changes that we made to the standard denoising objective were directly inspired by the DDPM framework. These changes contributed significant gains to the performance of our method. The discussion of the DDPM objective in the method description is therefore scientifically relevant, and we believe it can benefit the framing of the method.
>
> >”When the sample size of the downstream task increases, the advantages of the authors' method decrease.”
>
> It is true that the difference between supervised and DDeP pretraining decreases for large amounts of downstream data. We believe that this is expected, since the importance of any pretraining method decreases when enough fine-tuning data is available. Even the performance of from-scratch training improves substantially as downstream data increases. In the limit of infinite downstream data, we expect all methods to perform equally well. The advantage of our method lies in the regime where only a small amount of downstream data is available, which is often the case for semantic segmentation.
>
> >”If the training is long enough, even without pre-training, good results can be achieved.”
>
> This is a valid concern. We tested this by fine-tuning our models (both the “Supervised” baseline and DDeP) for up to 300 epochs. Even at the longest fine-tuning durations, DDeP pretraining still outperforms the baseline, showing that the effect of DDeP is not just to speed up training.
>
> The observations in [Rethinking ImageNet Pre-Training (https://openaccess.thecvf.com/content_ICCV_2019/papers/He_Rethinking_ImageNet_Pre-Training_ICCV_2019_paper.pdf) that models trained from scratch appear to perform competitively with fine-tuned models are potentially specific to the ImageNet 1K->COCO transfer setting. As that paper shows, fine-tuning still provides improvements over training from scratch on the smaller PASCAL VOC dataset. The target datasets we investigate are closer in size to PASCAL VOC than to COCO, and our models are also pretrained on datasets larger than ImageNet 1K.
>
> >Is the following statement in Section 3.4 wrong? "However, denoising pretraining of the full model underperforms standard supervised pretraining of the encoder, at least when fine-tuning data is abundant."
>
> This statement is correct. Please note the difference in our nomenclature between “Denoising Pretraining” (i.e. pretrain the entire model, both encoder and decoder, with denoising) and “Decoder Denoising Pretraining” (i.e. initialize the encoder with supervised weights and only pretrain the decoder using denoising).
>
> As shown in Table 2 (in the column titled “100%“), training the whole model with denoising performs worse than supervised initialization of the encoder. This motivated us to introduce Decoder Denoising Pretraining, which combines the advantages of supervised and denoising pretraining by initializing the encoder from supervised weights and training only the decoder using denoising.

---

> > ### Comment · Reviewer_B1kf · 2022-07-19
> > **follow up**
> >
> >
> > Before giving my final assessment, I would like to confirm with the authors whether the word "abundant" in the last question should be changed to "insufficient"?
> >
> > Best regards,

---

### Review · Reviewer_MKAe · 2022-06-13

**Summary Of Contributions:**

In this paper, the authors first point out a sub-optimal practice in the initialization of a dense prediction decoder. Typically, the most common approach to initialize a dense prediction model is to 1) inherit the pre-trained weights of the encoder from a large-scale image classifier and 2) randomly initialize the weights of the decoder. Then, the authors propose a denoising-based pretraining approach to initialize the decoder following the practice of denoising diffusion probabilistic models (DDPM).

Specifically, there are three steps in the proposed decoder denoising pretraining (DDeP) method. First, they initialize the encoder by large-scale image classification (i.e. ImageNet). Then, denoising pretraining is leveraged to initialize the decoder. Finally, both encoder and decoder are trained on semantic segmentation annotations.

**Requested Changes:**

- Visualize the reconstruction from step 2, a denoising-trained decoder
- Discussion on computational cost
- Improve the unfair comparisons, such as Table 5 and Table 6
- Explain why ImageNet-21K and why not ImageNet 1K
- More semantic segmentation methods with different encoders and decoders beyond TransUNet

Overall, I recommend accepting with major revision.

**Strengths And Weaknesses:**

### Strengths

1. **The proposed method is easy to follow.** The overall writing looks well-organized to me. Both big picture and implementation details are discussed intensively. I believe the readers could catch up with the most critical insights in an easy way.

2. **Both tables and illustrations are helpful** for the readers to realize the main motivations and contributions of the proposed method. One possible suggestion is to visualize the reconstructed images from a denoising-trained decoder.

### Weaknesses

1. **Missing discussion of the additional computational cost.** It is well known that DDPM requires a time-consuming training procedure. It could be slightly unfair to compare with a randomly initialized decoder since the total amount of FLOPs is quite different. Therefore, I am suggesting that the authors should at least mention the additional computational cost.
2. **The main comparison table (Table 5) has an unfair comparison.** DDeP leverages TransUNet as the backbone, which is quite different than the rest of the methods, which utilize ResNet-101. Both external dataset (ImageNet-1K vs ImageNet-21K) and backbone architecture (ResNet-101 vs TransUNet) are not the neglectable factors to have a fair comparison.
3. **The comparison on label-efficient semantic segmentation (Table 6) looks unfair to me.** Even though the total labeled images are limited to 1%, 5%, 10%, and 20%, the denoising training seems to work on the whole training set or external pretraining data (ImageNet-21K) with extra-long training epochs (100 epochs).
4. **More semantic segmentation methods are welcomed.** Since the proposed decoder pretraining seems to be proposed as a backbone-agnostic architecture, more semantic segmentation methods are welcomed, including HRNet, Semantic FPN, Segformer, and so on. The key idea is to confirm whether the lightweight decoder (asymmetrical arch) works or not.

---

> ### Author Response · Authors · 2022-07-07
> **Response to Reviewer MKAe.**
>
> Thank you for your valuable feedback! Below, we address the four weaknesses mentioned in your review. We have included the other requested images in the revised manuscript. See Figure 8 for a visualization of the reconstructed image in step 2 of the method.
>
> >”Missing discussion of the additional computational cost”
>
> Thank you. We agree that a discussion of computational costs of denoising pretraining is important. We have added a discussion to the revised manuscript. In summary, for our best model, the compute spent on decoder denoising pretraining is about 2 times the compute (FLOPS) used for supervised pretraining of the encoder.
>
> We acknowledge that this is a large amount of additional compute, but it is important to note that pretraining only has to be performed once, for all possible downstream use cases: Our results show that the same pretraining dataset (ImageNet 21K) transfers well to many downstream tasks across a wide range of downstream image types (i.e. Cityscapes, Pascal Context, and ADE20K). Therefore, pretraining cost has to be paid only once, and is then amortized over all downstream uses of the checkpoint.
>
> We plan to release our pretrained checkpoints, so the community will be able to benefit from our method without having to pay for the computational cost of pretraining again.
>
>
> >”The main comparison table (Table 5) has an unfair comparison.”
>
> It is true that Table 5 conflates pretraining and architecture differences. We have made this clearer in the revised manuscript. Our goal with Table 5 is to show that our method as a whole, including both pretraining and architecture, is competitive. We have updated the caption of Table 5 to state this more explicitly.
>
> While Table 5 contains a mix of architectures and training methods, many of the other analyses in our paper directly compare our Decoder Denoising method to carefully matched baselines, i.e. on the same architecture (e.g. Tables 2, 6, 7; Figure 6, 7). These results show that Decoder Denoising pretraining is beneficial across various datasets and architectures. It is therefore clear, independently of Table 5, that Decoder Denoising pretraining provides benefits that cannot be attributed to the TransUNet architecture alone.
>
> We now include an additional experiment (Table 8) with an ImageNet-1k-pretrained ViT backbone (instead of the Imagenet-21k backbone used in Table 5). DDeP continues to outperform the supervised baseline in all limited labeled data settings. However, the overall performance is lower than the ImageNet-21k pre-training. This is likely because ViT models require larger-scale supervised pretraining than ResNets. This model achieves 77.86 mIOU on full Cityscapes and outperforms 75.99 mIOU of PC2Seg (previous SoTA in Table 5).
>
> | Method         | Decoder width | full (2975) | 1/4 (744) | 1/8 (372) | 1/30 (100) |
> |----------------|----------------|-------------|-----------|-----------|------------|
> | No Pretraining | 1x             | 63.47       | 39.63     | 34.74     | 25.79      |
> | Supervised     | 1x             | 77.11       | 70.04     | 65.07     | 52.97      |
> | DDeP           | 1x             | 77.86       | 70.77     | 66.01     | 54.11      |
>
> >”The comparison on label-efficient semantic segmentation (Table 6) looks unfair to me.”
>
> We do not understand the concerns about Table 6. In Table 6, both the baseline (“Supervised”) and our method (“DDeP”) use the same exact dataset, and the same labels, for pretraining: In both conditions, the encoder is pretrained on ImageNet-21K using supervised classification. For DDeP, the decoder is additionally pretrained on ImageNet-21K images (without labels). In addition, both conditions use exactly the same data splits during fine-tuning on Pascal Context.
>
> It is true that DDeP requires additional computation compared to the baseline, for decoder pretraining. As described above, we will provide additional details about computational cost in the revised manuscript.
>
> >”More semantic segmentation methods are welcomed.”
>
> Please refer to Table 7 in the original manuscript, where we verify that DDeP works for a simple ResNet50 backbone, in addition to the TransUNet used in the remainder of the paper.
>
> We also would like to point to our experiments where we vary the decoder width (e.g. Table 6, Figures 6, 7), which show that our method works across a range of decoder sizes. While we agree that it would be interesting to try our method on other architectures, implementing each of them is a substantial effort. By making our code public, we will allow researchers to replicate our method on their architecture of choice.

---

### Review · Reviewer_3RdY · 2022-06-19

**Summary Of Contributions:**

#### Summary

They present a new method of pre-training a model for semantic segmentation, where they initialize an encoder with a model pre-trained with supervised learning and train a decoder with denoising loss while fixing the parameter of the encoder. Through experiments on various datasets, they show that the approach is label efficient pre-training approach for semantic segmentation.

#### Contributions
1. They propose a simple approach to pre-train the encoder-decoder architecture of semantic segmentation.
2. They perform various analyses in experiments to see the behavior of the approach, which shows convincing results.
3. Their approach sheds light on the importance of pre-training for the decoder network.

**Broader Impact Concerns:**

I do not see any ethical concerns in this submission.

**Requested Changes:**

1 and 2 in weaknesses are critical for my recommendation. 3 will just strengthen the work.

**Strengths And Weaknesses:**

### Strengths

1. A simple approach for the pre-training of semantic segmentation. Due to the simplicity of their approach, it can be effective for other types of datasets such as medical image segmentation.
2. Their experiments cover most of the contents to analyze the behavior of their approach.
3. They focus on the importance of a pre-training decoder network for semantic segmentation, which may be a new idea.

### Weaknesses
1. A major weakness is that they do not perform comparisons with other self-supervised learning approaches, which can be applicable to the decoder training such as inpainting or image colorization. If they state the importance of denoising-based self-supervised pre-training, they should present a comparison to these approaches. Also, the comparison to these baselines will reveal the effectiveness of denoising based on self-supervised learning.
2. I am curious why they choose to initialize an encoder with a supervised model. One natural option is initializing with a model trained with self-supervised approaches. Since most experiments are conducted on the supervised encoder, I am not sure whether their model shows superiority over encoders initialized differently.
3. Does the method work well on other types of networks such as ResNet? It will be great to add the results on different architectures.

---

> ### Author Response · Authors · 2022-07-07
> **Response to Reviewer 3RdY.**
>
> Thank you for your valuable feedback! Below, we address the weaknesses that you identified:
>
> >”No comparisons with other self-supervised learning approaches.”
>
> We agree that comparing denoising to other self-supervised approaches for decoder pretraining is a worthwhile research direction and may lead to further increases in performance. That being said, we believe that such comparisons are beyond the scope of this paper.
>
> The key innovation of the paper is that we separate pretraining of the decoder from pretraining of the encoder, which opens the door to a large number of combinations of techniques for decoder and encoder pretraining. In this paper we focus on denoising pretraining of the decoder, as it’s one of the simplest approaches. We perform extensive experiments to identify several key design choices of the denoising objective that are crucial for representation learning performance.
>
> We do not claim that denoising is the best method for decoder pretraining; our key claim is that denoising pretraining of the decoder significantly outperforms random initialization of the decoder. Drawing scientifically valid conclusions regarding the superiority of denoising pretraining over other decoder pretraining methods, e.g. inpainting and colorization, would require similarly extensive experiments to examine design choices for each of these methods as well. As anecdotal evidence, we tried both inpainting and colorization tasks in early experiments, and found that they both underperform denoising, despite being more complex. We therefore focused on denoising and left the study of other self-supervised decoder pretraining approaches to future work.
>
> We have included a discussion of the possibility of using other self-supervised methods in the revised manuscript. We have also briefly mentioned our preliminary findings concerning colorization and inpainting.
>
>
> >”Why initialize the encoder with a supervised model?”
>
> Initializing the encoder from a supervised model is a practical choice. Initializing models with weights trained with supervised classification is the standard approach in practice, since it is well-established and performs at or near state-of-the-art levels. While self-supervised methods are competitive on the ImageNet 1K dataset, they often underperform supervised pretraining on larger datasets (e.g. ImageNet 21K), and are more complex and difficult to implement. We therefore chose the simple and established option of supervised pretraining for the encoder.
>
> We do not claim that supervised pretraining of the encoder is necessarily the optimal choice. Instead, we claim that with the standard choice of supervised pretraining of the encoder, our denoising approach to pretraining the decoder is useful. We have amended the conclusion to acknowledge that we have only tried supervised pretraining for encoder initialization, and that other methods may be interesting to explore as part of future work.
>
>
> >”Does the method work well on other types of networks such as ResNet?”
>
> The generality of our method is an important question. Please refer to Table 7 of the original manuscript, where we have addressed this question for ResNets. As shown in Table 7, Decoder Denoising Pretraining outperforms supervised encoder pretraining across all label fractions for a model that is based on a ResNet50, instead of the TransUNet used in the rest of the paper. These results suggest that our method generalizes beyond transformer-based architectures to standard ResNet architectures.

---

### Author Response · Authors · 2022-07-07
**Common response to all reviewers.**

We thank the reviewers for their time and thoughtful comments. In particular, we thank the reviewers for their positive feedback on the simplicity of our method (3RdY, MKAe), helpfulness of analyses (3RdY), and the quality of the presentation (MKAe), and the overall importance of this line of work (B1kf:  **“... the idea is wonderful. This article is very meaningful.”**). We will respond to each reviewer separately in a direct response. We have  extensively modified the paper and included a discussion of reviewers’ comments. We believe the resulting manuscript is stronger and clearer.

---

### Comment · Reviewer_B1kf · 2022-07-20
**Post-rebuttal Comment**


After carefully reading colleagues' review comments and the authors' replies, I believe this paper has no major problems. Given that the significance of this paper fits well with the goals of the TMLR journal, I think this paper deserves acceptance.

The authors responded to most of my concerns. Thanks to the authors.

I understand that Reviewer 3RdY is concerned about the generalizability of this method using other self-supervised techniques, and Reviewer MKAe is worried about the generalizability of this method on other segmentation frameworks. As the authors replied, the article has enough experiments to prove the effectiveness of its method. Running experiments on these new frameworks is just icing on the cake, but not necessary to demonstrate the effectiveness of this approach.

I appreciate Reviewer 3RdY's concern about why the authors did not use self-supervision as initialization. An important driving force for the development of self-supervised techniques in recent days (whether in the CV field or in the NLP field) is to improve the knowledge reserve capacity of the model, that is, initialization. So it would be better if the authors could better respond to this concern of Reviewer 3RdY. Thanks to Reviewer 3RdY.

I appreciate Reviewer MKAe's concerns about the additional computational cost of our approach and the unfairness of Table 5. Fortunately, the authors responded reasonably. Thanks to Reviewer MKAe and the authors.

All in all, I recommend acceptance of this paper.

---

### Decision · Action_Editors · 2022-08-06

**Recommendation:** Accept as is

**Comment:**

The authors propose a new pre-training method semantic segmentation. In particular, rather than common pre-training which only focuses on the encoder (backbone), they find that decoder (head) pre-training is even more effective. Extensive experiments demonstrate the effectiveness of the proposed method. All three reviewers acknowledged the above merits of the paper and were concerned about the generalization ability and several technical details. The authors provided detailed feedback and improvements, which satisfied the reviewers.

AE recommends acceptance. Also, AE suggests that the authors may highlight some potential aspects of the method that has yet to be investigated.